# Disparate Environmental Monitoring as a Barrier to the Availability and Accessibility of Open Access Data on the Tidal Thames

**Julia Lanoue** 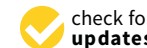

Thames Estuary Partnership, The Office of the Vice-Provost (Research), University College London, 26 Bedford Way, London WC1H 0AP, UK; data@thamesestuarypartnership.org

**Abstract:** Open Access data plays an increasingly important role in discussions of environmental issues. Limited availability or poor quality data can impede citizen participation in environmental dialogue, leading to their voices being undermined. This study assesses the quality of Open Access environmental data and barriers to its accessibility in the Thames Estuary. Data quality is assessed by its ability to track long-term trends in temperature, salinity, turbidity, and dissolved oxygen. The inconsistencies found in the data required analyses and careful interpretation beyond what would be expected of a citizen. The lack of clear documentation and centralized database acted as a major barrier to usability. A set of recommendations are produced for estuarine monitoring, including defining minimum standards for metadata, creating a centralized database for better quality control and accessibility, and developing flexible monitoring protocols that can incorporate new hypotheses and partnerships. The goal of the recommendations is to create monitoring which can encourage better science and wider participation in the natural environment.

**Keywords:** citizen science; data quality; estuary; monitoring; open data; Thames

## 1. Introduction

In an increasingly digital society, data is a key commodity [1–3]. Data is continuously being collected, processed, and analyzed across numerous sectors in increasingly higher volumes and with progressively more complex methods [1,2]. Simultaneously, with the rapid growth of the Internet and social media platforms, the scientific and political worlds are experiencing greater levels of public engagement [4,5]. As society becomes more interconnected, its citizens are calling for greater transparency and participation in the decision-making processes, giving rise to the "Open Access movement" [6,7]. The Open Access movement began in the nineties and continues to the present [1,8]. It calls for public access to the methods and data collected by government, industry, and academia, particularly those that feed into policy decisions [6,9,10]. Allowing open access to data, reports, and findings increases research visibility and encourages more participatory research [1,10,11]. Public engagement, especially in the natural sciences, is rapidly growing in sectors such as non-governmental organizations and charities through citizen science activities [7,8,11].

The scientific community has taken notice of the positive impacts of citizen science, such as the ability to collect large-scale and long-term data sets [4,7,8]. Presently, citizen science is increasingly being used as a tool by researchers, with the number of publications mentioning citizen science growing in the past ten years. This growth can be attributed to more funding initiatives that incorporate citizen

engagement, greater availability of low-cost technologies, and leveraging citizen participation to collect large-scale data [4,7,8]. Besides contributing to new research, citizen science has also been used to empower local communities and redirect research priorities and policy [1,12–14]. Presently, citizens are not just involved in data collection; their participation is expanding to data analysis and interpretation [11,14,15]. Groups of citizens concerned about environmental issues can now access the technologies and data necessary to develop their own analyses, draw their own conclusions, and share their findings on social media [6,7,11,15]. Environmental data, however, are often challenging to analyze because of the complexity of the measured systems and other factors with which citizens may be unfamiliar, such as seasonality, extreme weather events, and natural variation.

To increase transparency and trust in its policies and decisions, the European Union has embraced Open Access by creating a platform for research and data to be collated and shared [9,10,16]. The United Kingdom's data portals rank amongst the highest in the world in terms of ease of use and accessibility [16]. However, it is difficult to assess data quality because currently there is no centralized hub for quality control and standardization of environmental data. Often, it is difficult to create comprehensive repositories and complete documentation because these efforts involve resources, funding, and cooperation between government, industry, and academia [17,18]. This can lead to copious quantities of data but in disparate formats and scattered locations [15,19,20]. The complexity of environmental data, coupled with the lack of resources to regularly perform quality checks on large Open Data repositories, can lead to accidental misuse and misinterpretation [3,15].

This study assesses the availability and accessibility of Open Access data available on the tidal extent of the River Thames by investigating if long-term data, defined here as measurements taken consistently over time, exist and if they can be used to track basic physical and biochemical characteristics of the estuary. Business and property developers along the Thames often use Open Access data to calibrate models or establish pre-construction conditions for Environmental Impact Assessments [21]. These data are also used by academics for research projects [22–25] and the numerous active citizen science and conservation groups to develop new projects, such as the Thames Estuary Partnership and its partners [26–30]. Estuaries, however, are notoriously difficult to monitor due to their characteristic high levels of variability [31–33]. The availability and accessibility of Open Access data on the tidal Thames is investigated by:

1. Locating sources of environmental data to determine which data are publicly available, as there is no centralized data hub in place;
2. Identifying barriers and accessibility issues, such as ease of use and interpretation of the data;
3. Analyzing the collated data to assess its ability to act as a long-term data set and suitability to be used by a citizen.

## 2. Materials and Methods

### 2.1. Study Area

The tidal extent of the River Thames and Estuary, located in Southeast England, stretches 110 km from its westernmost limit in Teddington out into the Outer Estuary then the North Sea [34]. Estuaries, as the intersection between fresh water from rivers and salt water from the ocean, are highly dynamic environments characterized by gradients in temperature, salinity, dissolved oxygen, and turbidity [32,35]. Tidal action is a major driver of variation in those four characteristics [36,37]. Tidal action, coupled with freshwater inputs, also causes vertical and horizontal gradients along the estuary [37]. The constant flux of temperature, salinity, dissolved oxygen, and turbidity defines the biological structure of an estuary [35,38].

These four defining features of the estuary are predicted to shift in response to climate change [39–41]. The average water temperature of the estuary is predicted to increase [39]. Rainfall patterns are expected to

shift, leading to drier summers and extreme rain events in the winter [25,42,43]. Warming is also expected to impact large-scale ocean circulation and, therefore, tidal action in the Thames [39,40]. This, in turn, is predicted to impact the productivity of the estuary as introduced nutrients from freshwater inputs are dependent on rainfall and tidal action [36,37,44]. These changes over time can alter the structure of the estuary resulting in salt intrusion, fish deaths, and stratification in the water column [40,45].

*2.2. Collation*

The initial step in data collation was searching through the archives of various agencies and organizations known to collect and hold data on the tidal Thames. These sources were the Environment Agency (EA), Department for Environment, Food and Rural Affairs (Defra), Zoological Society of London (ZSL), Port Authority of London (PLA), Natural History Museum (NHM), London Database, Centre for Ecology and Hydrology (CEH), Medin, National Biodiversity Network (NBN), Greenspace Information on Greater London (GiGL), and HR Wallingford. If it was possible to explore the data, keywords on location such as "Thames", "London", and "estuary", were queried in addition to the four parameters "temperature", "salinity", "dissolved oxygen", and "turbidity". If the database could not be readily accessed, either the custodian was contacted if those data were available for release or the barriers to accessing the data on the website were noted.

*2.3. Analysis*

The Seasonal Mann–Kendall trend test was used to analyze each physical parameter using the Kendall package in R version 3.5.2 [46,47]. The test is a non-parametric modification of the Mann–Kendall trend test to specifically analyze environmental time series data, where strong seasonality and serial dependence are present [48–50]. The seasonal Mann–Kendall determines changes in mean values across time, a method used to reduce skewness caused by extreme values [48]. The output is a single test statistic $\tau$, a coefficient between $-1$ and 1 defined as the normalized summation of pairwise comparisons for measurements for each month within the data set [46]. The sign of $\tau$ indicates if a trend is positive, negative, or unchanging, and a statistically significant result indicates a non-random trend [50,51].

Spatially, the data were sorted into three categories: the estuary as a whole; location along the horizontal salinity gradient, referred to here as 'salinity category' (fresh, brackish, or marine); and general sampling site. The general sampling site is the specific site name at which samples were collected. The salinity categories are defined as the following: fresh waters have a salinity of below 0.5 ppt, brackish have between 0.5 and 30 ppt, and marine have over 30 ppt respectively [52]. For general sampling sites, only sites that had more than three years of data present were included in the analysis (Figure 1). In this study, three years is defined as the minimum amount of data needed to track a trend while still retaining the most amount of data possible (Figure 1). The Seasonal Mann–Kendall trend test was performed at each spatial scale.

For the statistical analyses, one major consideration was that the Thames is still the site of active ports, development, construction activities, and dredging, all of which lead to disturbances that can impact environmental measurements [21,53–56]. To mitigate any impacts from disturbance on the analyses, only sites and parameters with over ten years of data were selected. This decision was based on the results of previous studies in the Thames and other estuaries which noted that at least ten years of data were needed to track ecosystem recovery after disturbance [54,57,58] (Figure 1). Using this assumption, any short-term changes to parameter values due to construction or any other development activities should not impact the overall trends.

All dates, locations, and parameters were standardized to ensure consistency across the different data sets. Through standardization, it was possible to easily identify outliers and determine the highest

temporal resolution for the data. Outliers, defined here as values greater than two standard deviations from the mean, were removed prior to analysis as values outside this bracket are likely to be the result of measurement error [59].

The four parameters highlighted in this study—temperature, salinity, dissolved oxygen, and turbidity—can be reported in a variety of units. To capture as much data as possible, all units which can be used to measure those four parameters were included in the search. Salinity, for example, can be measured in parts per thousand (ppt), conductivity, or chlorinity. Dissolved oxygen can be reported in mg/L or percent saturation. Turbidity is commonly measured in Nephelometric Turbidity Units (NTU)/Formazin Turbidity Unit (FTU) or as suspended solids. NTU/FTU is a measure of the amount of light able to pass through a sample, and suspended solids record the dry weight in a sample heated to either 105 or 500 °C. Data on each unit were analyzed separately to determine trends were consistent between units of the same parameter. For example, measurements in ppt, conductivity, or chlorinity should report similar trends as they all represent salinity. The data were also checked for any accompanying tidal and sample depth information to ensure the sampling time relative to the stage in the tidal cycle was consistent across the data sets. This is important because of the variability that tidal action causes in the four parameters [36,37].

**Minimum Data Standards for Analysis**

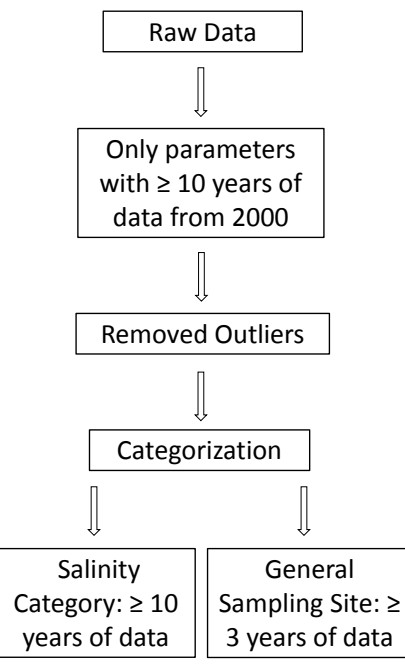

**Figure 1.** Flow chart outlining the data manipulation used to subset the data at each stage of analysis.

## 3. Results

### 3.1. Collation

Of the above searches, only three data sets from the EA archive contained readily accessible information on water temperature, salinity, dissolved oxygen, and turbidity on the tidal Thames. These data sets are the Water Information Management System (WIMS) [60], Automatic Quality Monitoring Systems (AQMS) tidal monitoring data (via link sent through email enquiry), and a data set provided from the EA through an email enquiry. All other sources disclosed that they either used the above EA

data for their environmental monitoring, the data was not available to the public due to industry-sensitive information, or the data focused exclusively on non-tidal portions of the Thames. Some data on the tidal Thames, such as those from HR Wallingford and GiGL, were behind a paywall or required paid membership for access.

The three relevant accessible data sets were all of differing time scales and resolutions. The EA queried data began in the 1980s until the present. The WIMS data started in 2000 [34,60]. The AQMS data only contained measurements for 12 months prior to the date of access. Data were most consistently available across all three data sets from 2000 to 2018. The WIMS and EA queried data provided the longest time series with at least monthly averages available. The AQMS data contained the highest resolution as measurements were collected every fifteen minutes (Table 1). There were no clear metadata on how and where the data are archived. Metadata for each of the data sets provided little to no information on instrumentation, collection methods, or details of the monitoring programs that contributed to each data set. This lead to some confusion, including not realizing that parts of the AQMS archives were incorporated as monthly averages within the EA queried data. It was not possible to locate a complete Open Access archive for the AQMS data. The data which the EA sent via the query also held little documentation. The data was sent via Dropbox with the open license agreement and a list of site names. It was also not possible to locate a name for the database from which it was extracted or the monitoring program by which it was collected. The data in the EA queried set did not overlap with the WIMS or AQMS data set, so no repeated measures were incorporated into the calculations. Despite repeated searches and queries, the source of the EA queried data remains unknown.

**Table 1.** Table describing the three data sets used in the analysis. For each data set, the time scale, parameter units, and temporal resolution is listed. The Environment Agency (EA) queried data had the longest time scale, followed by the Water Information Management Systems (WIMS) data sets. The Automatic Quality Monitoring Systems (AQMS) data set had the shortest time scale but had the highest resolution, with measurements being recorded every 15 min.

| Data Set | Time Range | Key Parameter Units | Resolution |
|---|---|---|---|
| EA queried | 1989–2018 | Temperature, salinity (ppt, conductivity), dissolved oxygen (mg/L, % saturation), turbidity (Nephelometric Turbidity Units (NTU)/Formazin Turbidity Unit (FTU), suspended solids). | Daily, weekly, or monthly |
| WIMS | 2000–2018 | Temperature, salinity (ppt, conductivity), dissolved oxygen (mg/L, % saturation), turbidity (NTU/FTU, suspended solids). | Weekly |
| AQMS | 2017–2018 | Temperature, salinity (ppt, conductivity), dissolved oxygen (mg/L), turbidity (NTU/FTU). | Fifteen minute intervals |

After cleaning the data to ensure at least ten years of data were present, all four parameters retained enough information for further analysis and were reported in the data sets using seven different units: temperature ($^\circ$C), salinity (ppt), conductivity at 25 $^\circ$C (S/m), dissolved oxygen (mg/L and % saturation), turbidity (NTU/FTU), and suspended solids at 105 $^\circ$C (mg/L).

Information on tides and sample depth was also sparse. Tidal information was available for about 25% of the data, where it was presented as the time at high tide. Tidal information was only present in the data from 2010 onwards. Sample depth information was only available for about a third of the data and was recorded from 2000 onwards. No data points contained both tidal and sample depth information. Most measurements were taken close to high tide, but there were a number of measurements that were

taken at other times during the tidal cycle. Sample depth also varied across the estuary, with sample depths ranging from 0 to 27.8 m. The average depth was 0.56 m, with the median sample depth at 0.20 m.

### 3.2. Seasonal Mann–Kendall

The collated data revealed incomplete, and at times contradictory, information between data sets, including a number of negative and physically improbable values for the Thames, such as recorded water temperatures of over 100 °C and salinity values of over 70 ppt. The level of fragmentation in the data was also reflected in the results of the Seasonal Mann–Kendall. At the estuary level, no strong trends were identified ($|\tau| < 0.5$). There were four statistically significant trends: salinity, suspended solids at 105 °C, temperature, and turbidity (Figure 2, Table 2). The trends in salinity (Figure 2B), suspended solids (Figure 2C), and turbidity (Figure 2D) are visually striking, showing drastic increases or decreases over time, while the slight decrease in temperature identified in the Seasonal Mann–Kendall cannot be easily seen (Figure 2A). Trends in dissolved oxygen (mg/L and % saturation) (Figure A1B,C) and conductivity (S/m) (Figure A1A) were not statistically significant.

**Whole estuary time series**

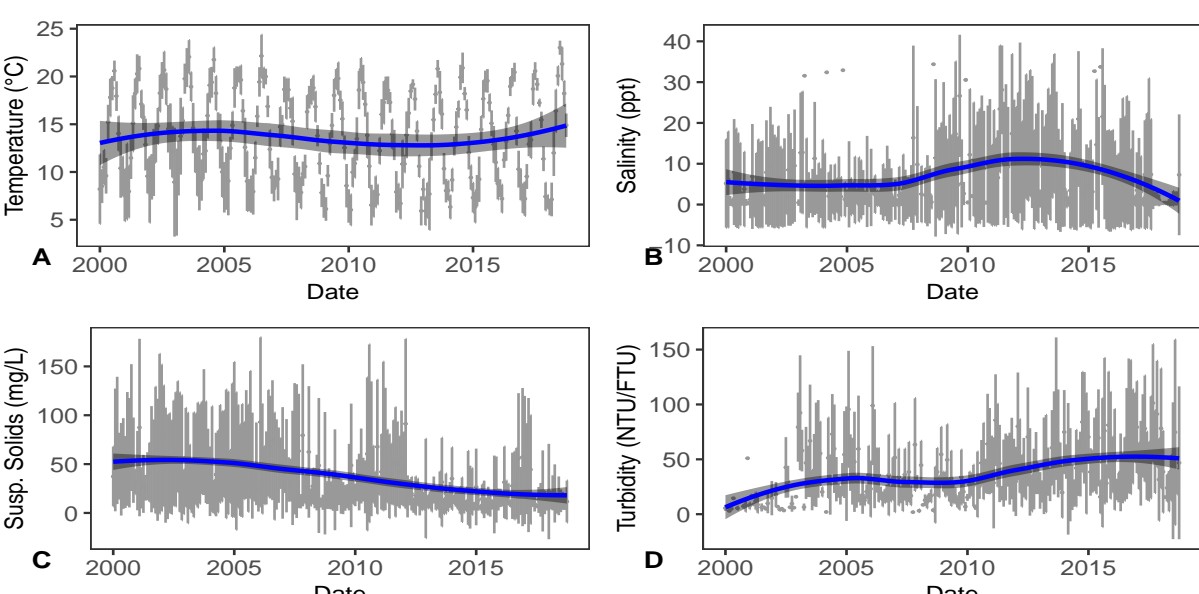

**Figure 2.** Monthly averages with standard deviations for each statistically significant parameter unit across time. The blue line represents a non-linear regression line. The results of the Seasonal Mann–Kendall showed no strong trends. Temperature (**A**), salinity (**B**), suspended solids (**C**), and turbidity (**D**) showed statistically significant trends. The trends in salinity, suspended solids, and turbidity are easily seen, while temperature displays a more cyclic trend.

**Table 2.** Table outlining the results of the Seasonal Mann–Kendall for parameters at the whole estuary level. The only significant trends were temperature, salinity, suspended solids, and turbidity (Figure 2). Both units for dissolved oxygen and conductivity did not reveal significant trends (Figure A1).

| Parameter | $\tau$ | *p*-Value |
|---|---|---|
| Temperature (°C) | −0.247 | $5.19 \times 10^{-7}$ |
| Salinity (ppt) | 0.104 | 0.0352 |
| Suspended Solids (mg/L) | −0.457 | $<2.22 \times 10^{-6}$ |
| Turbidity (NTU/FTU) | 0.328 | $2.80 \times 10^{-11}$ |
| Conductivity (S/m) | −0.0587 | 0.234 |
| Dissolved Oxygen (mg/L) | 0.0388 | 0.431 |
| Dissolved Oxygen (% sat) | −0.021 | 0.671 |

Grouping the data by salinity category revealed the locations along the Thames where statistically significant trends originated as well as the data gaps which may be causing the trends. Results from the Seasonal Mann–Kendall for each salinity category also showed no strong trends. Conductivity and dissolved oxygen did not have statistically significant results across all categories (Figure A2). The Seasonal Mann–Kendall did reveal statistically significant trends in salinity, suspended solids, temperature, and turbidity (Figure 3, Table 3). During the analysis at this level, changes in sample size over time were also considered. Overall, the number of samples collected in each parameter has greatly decreased over time, with often fewer than 100 samples collected per year after 2007–2008 (Figures 3 and A2).

**Table 3.** Table summarizing the results of the Seasonal Mann–Kendall for each unit by salinity category. The only statistically significant trends are in temperature (brackish and marine), salinity (brackish), suspended solids (fresh, brackish, and marine), and turbidity (brackish). All other trends are not statistically significant.

| Parameter | Salinity Category | $\tau$ | *p*-Value |
|---|---|---|---|
| | Fresh | 0.053 | 0.278 |
| Temperature (°C) | Brackish | −0.271 | $3.63 \times 10^{-8}$ |
| | Marine | −0.171 | $5.27 \times 10^{-4}$ |
| | Fresh | 0.034 | 0.496 |
| Salinity (ppt) | Brackish | 0.145 | $3.34 \times 10^{-3}$ |
| | Marine | 0.045 | 0.360 |
| | Fresh | −0.155 | $1.65 \times 10^{-3}$ |
| Suspended Solids (mg/L) | Brackish | −0.420 | $<0.001$ |
| | Marine | −0.402 | $2.22 \times 10^{-16}$ |
| | Fresh | 0.039 | 0.425 |
| Turbidity (NTU/FTU) | Brackish | 0.276 | $2.23 \times 10^{-8}$ |
| | Marine | −0.0503 | 0.307 |
| | Fresh | 0.0363 | 0.468 |
| Conductivity (S/m) | Brackish | 0.0210 | 0.671 |
| | Marine | 0.00238 | 0.963 |
| | Fresh | −0.0314 | 0.523 |
| Dissolved Oxygen (mg/L) | Brackish | 0.0608 | 0.217 |
| | Marine | −0.0514 | 0.297 |
| | Fresh | 0.0126 | 0.799 |
| Dissolved Oxygen (% sat) | Brackish | −0.0367 | 0.457 |
| | Marine | −0.0288 | 0.559 |

**Trends by salinity category and corresponding sample sizes**

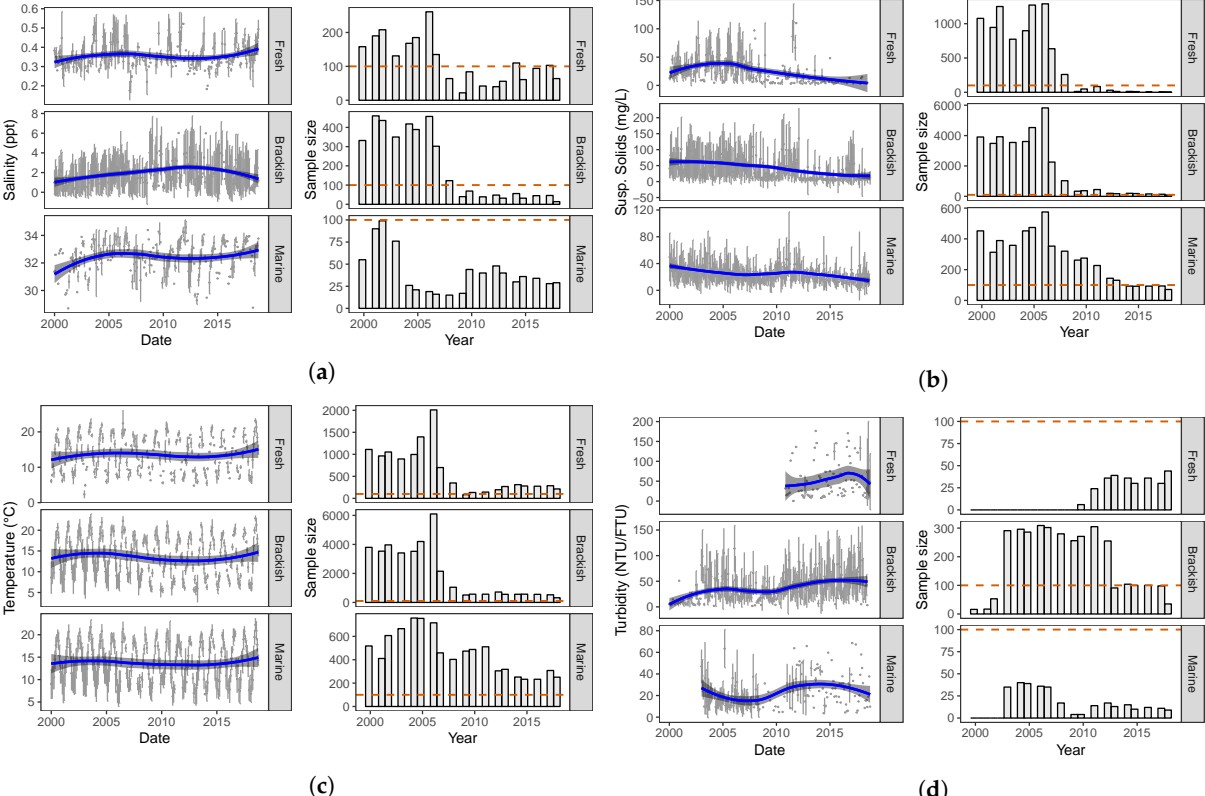

(**a**)　　　　　　　　　　　　　　　　　(**b**)

(**c**)　　　　　　　　　　　　　　　　　(**d**)

**Figure 3.** Trends for each statistically significant unit over time with accompanying histograms tracking sample size over time. The blue line represents a non-linear Loess regression to visualize trends, and the orange dashed line marks a sample size of 100. Data from the AQMS 12 month data set was not included in the counts because the high resolution level in the AQMS data is atypical compared to the data available outside that 12 month time period. Statistically significant but weak trends were noted in salinity (**a**) in the brackish category, suspended solids (**b**) in all categories, temperature (**c**) in the brackish and marine categories, and turbidity (**d**) in the brackish category.

Analysis of the general sampling sites for each parameter yielded 210 results. None of the results showed strong trends. There were ten results which were statistically significant with weak trends: dissolved oxygen in mg/L at Mucking, salinity at Greenwich, temperature at Gravesend, and suspended solids at Basildon, Canvey Island, Crossness, Dartford, Greenhithe, Pitsea, and Wandsworth (Table 4). Besides dissolved oxygen, where the sample size has remained relatively constant across time (Figure 4a), all other statistically significant trends also showed a decrease in sampling effort. The year at which sample size decreased frequently coincided with drastic shifts in the average parameter values (Figures 4 and A3).

**Table 4.** Table summarising the statistically significant results of the Seasonal Mann–Kendall trend test for the general sampling sites.

| Parameter | General Sampling Site | $\tau$ | *p*-Value |
|---|---|---|---|
| Dissolved Oxygen (mg/L) | Mucking | 0.105 | 0.0334 |
| Salinity (ppt) | Greenwich | 0.180 | 0.0268 |
| Temperature (°C) | Gravesend | −0.126 | 0.0107 |
| | Basildon | 0.115 | 0.0211 |
| | Canvey Island | −0.217 | $1.24 \times 10^{-5}$ |
| | Crossness | −0.288 | $5.76 \times 10^{-9}$ |
| | Dartford | −0.298 | $1.54 \times 10^{-9}$ |
| Suspended Solids (mg/L) | Greenhithe | −0.164 | $8.98 \times 10^{-4}$ |
| | Pitsea | −0.129 | $9.32 \times 10^{-3}$ |
| | Wandsworth | −0.130 | $8.34 \times 10^{-3}$ |

### Trends in general sampling sites and corresponding sample sizes

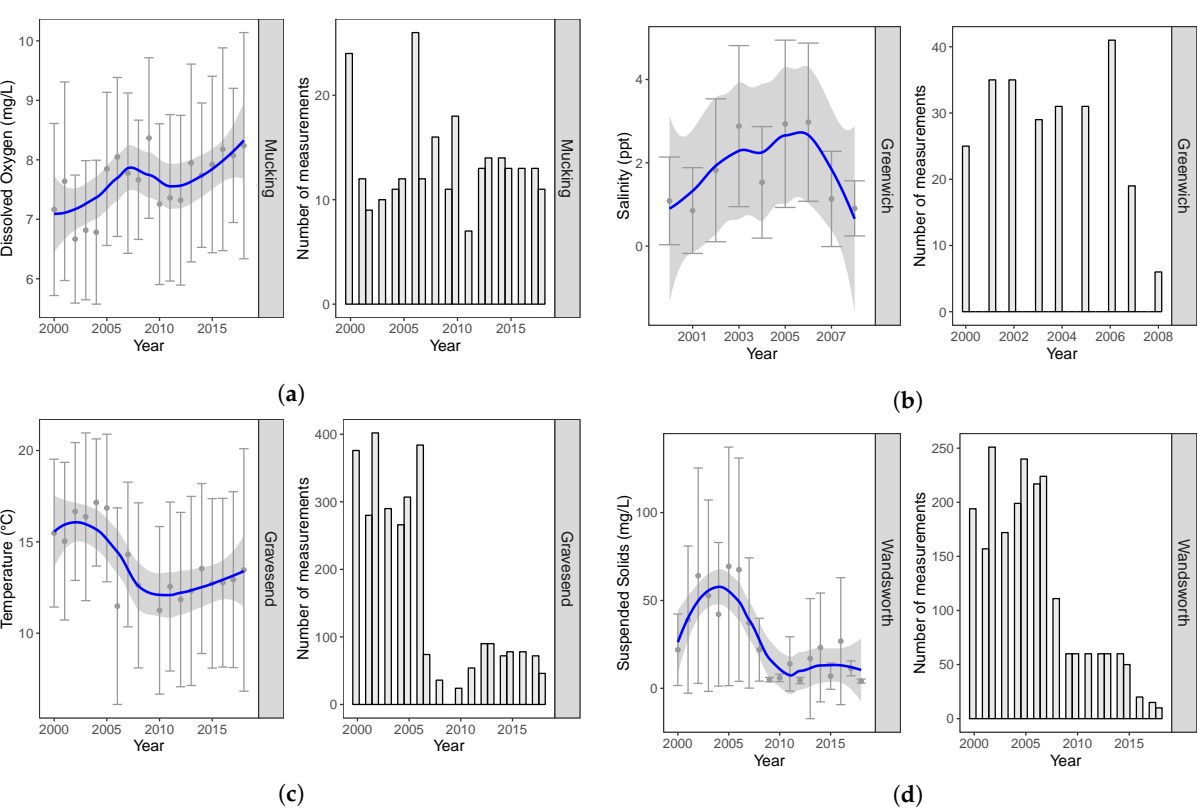

**Figure 4.** The general sampling sites with weak but statistically significant trends over time. None of the analyzed sites displayed strong trends. (**a**) Dissolved oxygen levels over time at Mucking. This is the only parameter where sample size remained constant. The trend showed a gradual increase in dissolved oxygen levels over time. (**b**) Salinity levels at Gravesend over time. In the last two years, the amount of samples available drastically decreased. The salinity averages at the site also decrease at this time. (**c**) Water temperature levels over time. The Seasonal Mann–Kendall noted a statistically significant decrease in temperature. The drop in average temperature values coincides with a dramatic drop in sample size over time. (**d**) Subset of results for suspended solids over time. The changes in suspended solid concentrations over time coincide with decreases in sampling. Figures for each general sampling site for suspended solids can be seen in Figure A3.

## 4. Discussion

The River Thames is a London icon, representative of culture, industry, and more recently a conservation success story [34,56]. Disjointed restoration and monitoring efforts have scattered data across numerous organizations, and lack of communication between these efforts has caused confusion on the location and ownership of data on the Thames [53]. In an increasingly data-centric society, transparency and accessibility to data plays a major role in citizen engagement and empowerment, where greater access to the Internet, social media, and low-cost technologies allows citizens to independently collect, analyze, and interpret data [7,8,11]. For public and private bodies that hold environmental data, it is essential to provide Open Access, clear instructions on data use, clear and concise metadata, and high-quality data to maintain transparency. This study investigated the availability and accessibility of environmental data on the tidal Thames to determine if any long-term data existed and if it could be used to track change. A large amount of the data on the tidal Thames are rendered unavailable through a paywall or deemed too sensitive for public release. As a result, numerous organizations rely on the same few data sets from the Environment Agency for their analyses. However, the data greatly lack relevant, concise documentation and quality control, creating barriers to the credibility and usability of the data.

Estuaries are productive, dynamic environments characterized by gradients in temperature, salinity, dissolved oxygen, and turbidity. However, the structures of estuaries are threatened because of climate change. The expected changes include decreases in freshwater inputs [40], increased water temperatures [39,41], and stronger stratification in the water column [40,45]. Given the expected impacts, the minimal amount of information available on the tidal Thames is surprising. The majority of the organizations which monitor environmental parameters on the Thames focused solely on the North Sea or non-tidal freshwater sections of the Thames, overlooking the brackish section of the river. Those that did monitor the tidal Thames focused exclusively on biological or chemical parameters, such as fish, vegetation, invertebrates, or chemical pollutants. Though monitoring these aspects is important, these biological and chemical components respond to changes in the physical environment [35,38,61,62]. The organizations that do hold environmental data often store it in membership-only databases, like GiGL, or keep it behind closed doors at private consultancy firms. As climate change impacts the diversity and distribution of flora, fauna, and habitat, the lack of environmental data will hinder further studies to track the underlying mechanisms driving the change [63,64]. Overall, there is a severe lack of long-term data available to the public on the tidal Thames.

Of the data that were publicly accessible, none were comprehensive. There was minimal information available in the metadata or the database websites on information such as the monitoring programs under which the data was collected, the standards to which the data was measured, the longevity of the data, sample sites, and instrumentation. The lack of metadata is a major hindrance to data accessibility to the general public and increases the likelihood of misinterpretation [7]. The analyses at every level of the Thames showed that decreases in sampling frequency were the main driver for the statistically significant trends (Figures 3 and 4). Taken at face value, those trends could be used to naively justify erroneous claims pertaining to climate change or the health of the Thames. To prevent misuse, it is essential that data quality is monitored and the data are presented in a clear, concise, and standardized manner. The Environment Agency's database is also highly decentralized, with data sets stored across a number of different websites based on the monitoring program used for collection, the type of data, and the sampling location. The extensive searching and data standardization necessary to locate and extract the appropriate data can be a barrier to usability.

The decrease in sample sizes seen at every spatial scale can be traced to a policy change that shifted monitoring protocols. In 2006, a new monitoring program mandated through the Water Framework Directive (WFD) was implemented in the United Kingdom [65–67]. The WFD is a European directive

aimed at restoring water bodies to "good ecological status." This involves monitoring biological, chemical, and morphological elements and comparing their measurements to a reference value [67,68]. Implementation of the WFD involved monitoring a greater number of waterbodies with new additional parameters [67]. To make monitoring more cost effective, the number of sampling sites and frequency of sampling at each site decreased [64,69]. Some studies have already noted the impacts of decreased sampling on uncertainty levels in the measurements [70,71]. Though the WFD does acknowledge that temperature, salinity, and turbidity greatly contribute to the biological structure, monitoring of ecological status through the WFD focuses on flora and fauna [38,72]. The inconsistency in data resulting from this policy change requires data cleaning beyond what a citizen scientist would be expected to know, such as selectively matching samples taken pre-WFD and post-WFD based on tidal cycles, locations, and depth. Decreasing the sample size as drastically as seen in Figures 3 and 4 decreases the number of data points used to calculate the variance of a measurement. Estuaries are highly variable environments, and low sampling frequency and inconsistencies in sampling times can artificially cause trends, as seen throughout the analyses.

Four recommendations have emerged for future monitoring efforts on estuaries in order to produce high quality, Open Access data:

1. Metadata must have defined minimum standards. These standards should include information on instrumentation, tidal cycles, sampling depth, and background on the monitoring program.
2. Data should be stored in a centralized data system to increase availability and accessibility, promote standardization and quality control, and provide a standard which can be applied nationally. Examples of this data structure can be seen in the National Estuarine Research Reserve (NERR) system in the United States [73,74] and the South African Environmental Observation Network (SAEON) in South Africa [18,75]. In both programs, centralized databases are established in conjunction with national monitoring programs to ensure data are collected consistently and the database is maintained by frequent checks for quality control.
3. Basic physical parameters, such as temperature, salinity, dissolved oxygen, and suspended solids, should be consistently monitored [38]. These parameters have been ignored in monitoring on the Thames, creating a data gap which must be filled as these parameters influence the structure and function of the Thames.
4. Future monitoring programs should be guided by question-driven study design. Question-driven designs often involve the creation of a conceptual model which can lead to the formation of hypotheses. This can, in turn, lead to a better understanding of ecosystems through hypothesis-testing instead of only accumulating data points. This opens opportunities for additional research questions and encourages partnerships between other organizations [17].

In an era of constant information flow and low-cost technologies, data accessibility and transparency play a major role in the interaction between citizens and larger organizations [4,8,11]. It is essential that organizations are transparent about the data available, techniques used, and interpretation of results. In the face of climate change, this is especially important for concerned citizens who want to be involved in protecting and learning about their local ecosystems. Estuaries are difficult to monitor because of their dynamic nature. The inconsistencies introduced by incomplete metadata, decentralized data hubs, and passive monitoring result in poor data. This can exclude citizens and local groups from conversations surrounding the environment, leading to feelings of distrust and disenfranchisement. The above recommendations aim to provide guidance for future estuarine monitoring to produce high quality, Open Access data in the hope of increasing dialogue across sectors, creating better science, and making science more accessible to wider audiences.

## 5. Conclusions

This study assessed the availability and accessibility of Open Access environmental data on the tidal Thames. Availability is defined as the ease of locating data, and accessibility is the ease of data use and interpretation. An investigation into the available Open Access data sources revealed that only one data source is freely available for public use. Other data sources either do not release their data to the public or do not monitor the brackish portion of the tidal Thames. The data that were freely available were not accessible because of minimal information in the metadata. Lack of relevant metadata can be a major hindrance to its ease of use and interpretation. In this study, the metadata did not include information on a change in monitoring protocol that impacted sampling frequency and the number of sampling sites. The inconsistencies caused by the change in monitoring impacted the usability of the data as a long-term data set and its accessibility to citizen scientists. Based on the results, four recommendations were developed to improve estuarine monitoring as well as promote availability and accessibility of environmental data on the tidal Thames. The recommendations are the creation of minimum data standards for metadata, the development of a centralized data system to promote data availability and standardization, monitoring that consistently measures basic parameters, and the introduction of question-based monitoring to encourage collaboration and research. The availability and accessibility of Open Access data play a major role in the discourse surrounding environmental issues and public engagement. It is essential that data availability and accessibility improve to promote transparency, discourse, and inclusivity in environmental issues.

**Funding:** This paper is based on a project funded by the North Thames Fisheries Local Action Group, which is jointly supported through the Department for Environment, Food and Rural Affairs (Defra) and the European Maritime and Fisheries Fund (EMFF) under project reference number ENG2651.

**Acknowledgments:** The author would like to acknowledge Helen Czerski for her insightful feedback and comments, Chris Hampson for reading countless drafts and providing valuable feedback, Mark Davison and Neil Dunlop from the Environment Agency for their help in locating some data as well as insight into policy changes, and NTFLAG, Anna Patel, and Amy Pryor for funding.

**Conflicts of Interest:** The author declares no conflict of interest.

## Appendix A

In the analyses for the estuary as a whole, three parameters did not have statistically significant trends: dissolved oxygen (mg/L), dissolved oxygen (% saturation), conductivity (S/m) (Figure A1, Table 2). The non-significant results indicate that the trend cannot be determined as non-random. Measurements for conductivity are sporadic across time, with little consistency in sampling. Levels of dissolved oxygen remain relatively constant across time. The $\tau$ values are all very close to 0, indicating very weak trends.

Analyses by salinity category also showed non-significant, weak trends (Figure A2). For dissolved oxygen (mg/L), the results of the Seasonal Mann–Kendall are as follows: fresh: $\tau = -0.0314$, $p = 0.523$, brackish: $\tau = 0.0608$, $p = 0.217$, marine: $\tau = -0.0514$, $p = 0.297$. The results for dissolved oxygen (% saturation) are: fresh: $\tau = 0.0126$, $p = 0.799$, brackish: $\tau = -0.0367$, $p = 0.457$, marine: $\tau = -0.0288$, $p = 0.559$. The results of the Seasonal Mann−Kendall for conductivity (S/m) are: fresh: $\tau = 0.0363$, $p = 0.468$, brackish: $\tau = 0.0210$, $p = 0.671$, marine: $\tau = 0.00238$, $p = 0.963$. The $\tau$ values were all again very close to zero, indicating that the trends could not be discerned from random trends. For conductivity, the histograms tracking the number of samples per year visualize the sporadic sampling hinted at in Figure A2a. In the marine category, only eight years of samples were available. The most samples were collected in the brackish category, though values were unavailable for all years from 2000 to 2018. The only measurements available in the fresh water category originated from the AQMS data set, which was not included in the histogram count because the number of data points available is not reflective of the samples historically available. For dissolved oxygen (mg/L and % saturation), there was a decrease in the number of samples taken

across time. However, there are still more samples taken per year in dissolved oxygen then the other parameters (Figure A2b). Though there was a decrease in the number of samples in dissolved oxygen (mg/L), there were still over 100 samples collected each year in each category. For dissolved oxygen in % saturation, there has been consistently few samples taken per year (Figure A2c). The only category which experienced a sharp decrease is the brackish category, where less than ten samples were taken for four years.

### Whole estuary time series: non-significant parameters

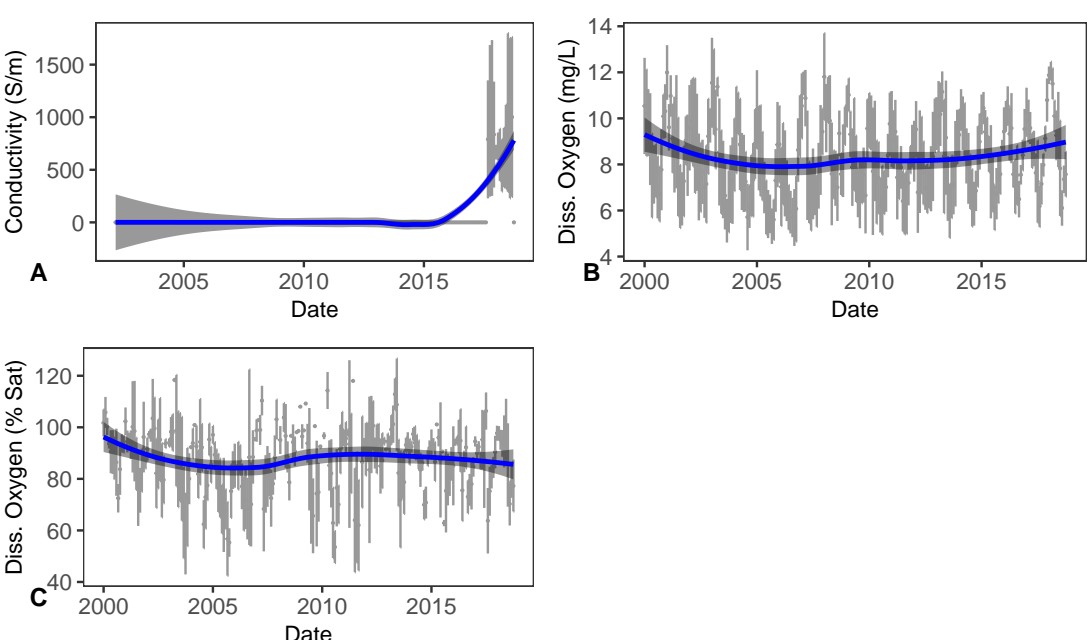

**Figure A1.** Non-significant trends from the Seasonal Mann–Kendall at the whole estuary analyses. (**A**) Conductivity measurements along the Thames from 2000 to the present. Data were sporadically collected, resulting in the statistical result which indicates the pattern cannot be identified as non-random. (**B**) Dissolved oxygen (mg/L) across time for the tidal Thames. The non-significant result from the Seasonal Mann–Kendall indicate the trend is not different from random variation. (**C**) Dissolved oxygen (% saturation) across time for the tidal Thames. Levels seem to remain constant over time, with the result of the Seasonal Mann–Kendall indicating that the trend could not be distinguished from natural variation.

**Trends by salinity category and corresponding sample sizes: non-significant parameters**

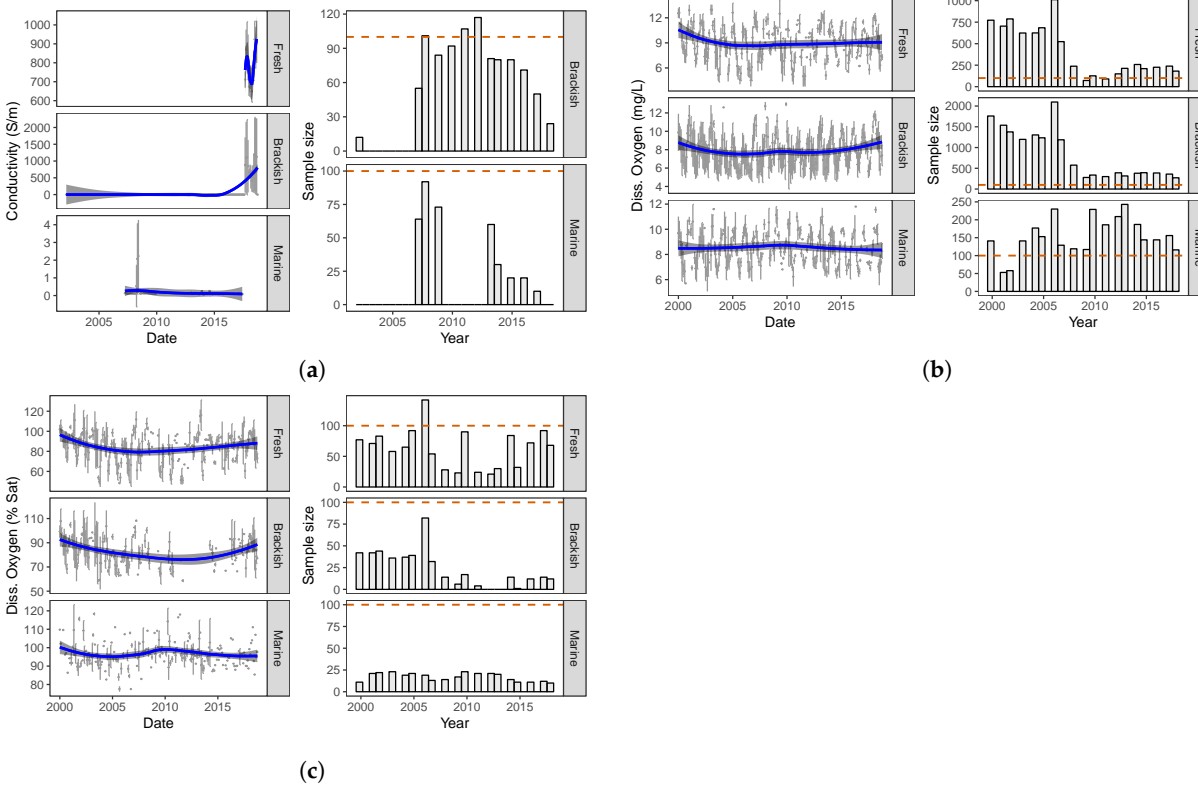

**Figure A2.** Non-significant trends and sample sizes for salinity category-level analyses. (**a**) Conductivity trends over time and sample sizes. Few conductivity measures are available over time. This low amount of data is reflected in the results of the Seasonal Mann–Kendall, which cannot distinguish the current trend from random variation. (**b**) Dissolved oxygen (mg/L) trends over time. Results from the Seasonal Mann–Kendall determined that the trend is not significantly different from natural variation, though the number of samples taken over time has greatly decreased in all categories except marine. (**c**) Dissolved oxygen (% saturation) trends and sample size over time. None of the trends were statistically significant. This may be due to the low number of samples taken per year.

Results of the Seasonal Mann–Kendall at the level of general sampling site showed significant but weak trends at Basildon ($\tau = 0.115$, $p = 0.0211$), Canvey Island ($\tau = -0.217$, $p = 1.24 \times 10^{-5}$), Crossness ($\tau = -0.288$, $5.76 \times 10^{-9}$), Dartford ($\tau = -0.298$, $p = 1.54 \times 10^{-9}$), Greenhithe ($\tau = -0.164$, $p = 8.98 \times 10^{-4}$), Pitsea ($\tau = -0.129$, $p = 9.32 \times 10^{-3}$), and Wandsworth ($\tau = -0.130$, $p = 8.34 \times 10^{-3}$) (Figure A3). Each site showed a decrease in the number of samples collected per year. With the exception of Basildon, the inflection in average values across time noted in each site coincided with decreases in samples. At Basildon, the decrease in samples coincided with an increase in the average suspended solids.

## Trends in suspended solids concentrations by general sampling sites

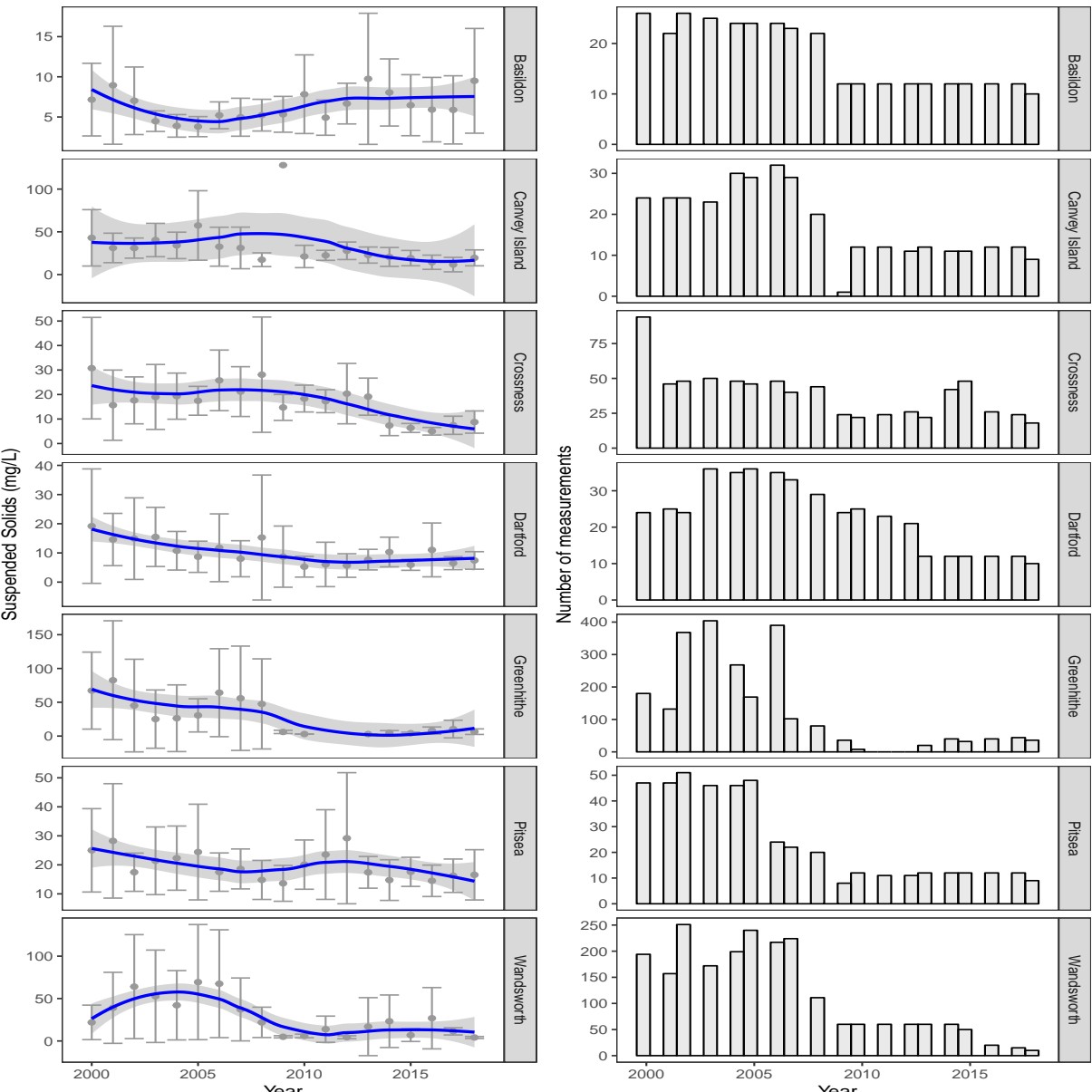

**Figure A3.** Trends and sample sizes over time for suspended solids (mg/L) at the general sampling sites. Each site showed a weak but statistically significant trend. The number of samples taken per year decreased at each site from 2000 to 2018. With the exception of Barking, all the sites showed a decrease in the average suspended solids value which coincided with a drop in samples taken per year.

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
