# Peer review of "Disparate Environmental Monitoring as a Barrier to the Availability and Accessibility of Open Access Data on the Tidal Thames"

_publications, doi:10.3390/publications8010006_

Round 1

Reviewer 1 Report

Overall, while I found this article very interesting and a highlight to the issues on data collected and maintained through disparate locations and agencies, it’s very hard to read in its current structure. While the author's findings and recommendations are sound, and provide an example of a disciplinary issue that has broader implications for the general public and the citizen scholar/researcher, the sections need to be reordered and the content of each section reviewed for potential repetition. The authors should review the Publications Microsoft Word template file and follow the structure provided in the template, MDPI style guide, and MDPI references guide. 

With revisions and changes by the below-listed comments, I believe this article will be useful to those wanting to understand both disciplinary data on the tidal Thames, but also on the issues and challenges facing citizen research and Open Data.

Overall Comments:

The tense should be reviewed. Most of the writing point of view is in the 1st person (e.g. I…), would be a stronger article if written and maintained in the 3rd person (e.g., The authors…., Those searching for Open Access datasets on tidal Thames…) Review the reference list for MDPI journals. many of the references do not include a date accessed when reviewing online materials or publications. “Section 4. Materials and Methods” should be Section 2. “Section 2. Results” should be Section 3. “Section3. Discussion” should be Section 4.

Individual By-Line Comments: 

20: Review if Open Access Movement should be capitalized as it refers to Open Access. Should be defined in the MDPI journal style guide.

24-26: Sentence “Domains where public engagement…” reads weird. Review inclusion of “..,are citizen science,..” seems out of place or not completed.

46: Comma between data and coupled – “data, coupled”

47: Comma between repositories and can – “repositories, can”

49: Define “Long-term data”, also include hyphen between long-term

58: Cite and reference projects related to Zoological Society of London and Thames21

69: Spell out Environmental Agency in first use, not done until line 152

72-75: Remove hyperlinks, include as citations within references

90: Remove “a” before “Dropbox”

109: include comma – “improbable values (,) such as…”

138-139: There is no caption or figure citation to “Trends in general sampling sites and corresponding sample sizes”

147: Review MDPI journal style guide to capitalize Open Access

Figure 3: lacks title, should be placed after figure identified as “Trends in general sampling sites and corresponding sample sizes”

161: Include “The” to begin sentence. “The majority of the organizations…”

164: Delete “and” as it is not needed

167: Include “in” - … “often sore it in membership only…”

Reviewer 2 Report

Dear author,

The subject is very interesting, and I think this is clearly a highly topical issue of major importance. However, before to be publish I would like to suggest various modifications to improve the comprehension of the article. For my point of view the author must restructure the article and to define more precisely the objectives.

At the end of the introduction it would be good to expose more clearly the objectives (lines 63-65) and to describe the nomenclature of the paper. Before to expose the results (line 67) it would be better to describe the Area of interest (part 2) and then the data and methodology (part 3), and only after that part 4 with results. The author must present the area of interest (maybe a map to indicate where is it, measure points, tidal part, selected sites, …). Line 69: must put the entire name of EA before the acronyms. Lines 79 – 94: resume the dataset in a Table to improve the comprehension (start date, end date, length, resolution, parameters, ...) Line 95 the authors write “only seven parameters” but only six are exposed (temperature, salinity, conductivity, dissolved oxygen, turbidity and suspended solid). It would be good if each parameter used in the study is resume in a Table. Line 107 the part Analysis looks more like results. And Put all your results in a Table. Figure 1 Where are the other parameters? At line 236 “Materials and methods” must be right after the introduction. Line 271: Now the authors speak about 4 parameters where are the two others? The conclusion must be more in line with the objectives beacause I don't really understant the principal results.

Round 2

Reviewer 1 Report

The author has addressed my previous comments. I believe this new version is stronger than the previous. 

Reviewer 2 Report

First of all I would like to thank the author for her great work and for taking forward my suggestions. I think that the manuscript is now much better to understand!

Before publishing it I just have some small remarks:

Line 29 Do you have an idea of the growth? Line 119 which common format? In the Discussion I really like the fact that the author gives some recommendations. It would be good if the author summarized all objectives in a small paragraph. There is no conclusion.
